# First Case of a Cerebrocortical Ganglioglioma in a Dog

**DOI:** 10.3390/vetsci9100514

**Published:** 2022-09-21

**Authors:** Laura Martín, Martí Pumarola, Raúl Altuzarra, Javier Espinosa, María Ortega

**Affiliations:** 1Neurology and Neurosurgery Service of Centro Clínico Veterinario Indautxu, Alameda de San Mamés 38, 48010 Bilbao, Spain; 2Mouse and Comparative Pathology Unit, Department of Animal Medicine and Surgery, Veterinary Faculty, Networking Research Center on Bioengineering, Biomaterials and Nanomedicine (CIBER-BBN), Campus UAB, Universitat Autònoma de Barcelona, Bellaterra, 08193 Barcelona, Spain; 3Diagnostic Imaging Service of Centro Clínico Veterinario Indautxu, Alameda de San Mamés 38, 48010 Bilbao, Spain; 4Neurology and Neurosurgery Service, Pride Veterinary Centre, Derby DE248HX, UK

**Keywords:** tumor, neoplasia, central nervous system, canine, glial fibrillary acidic protein, neuron-specific enolase, synaptophysin, ganglioglioma

## Abstract

**Simple Summary:**

Gangliogliomas are extremely rare tumors of the nervous system that contain a mixture of neoplastic glial and neuronal cells. The aim of the present paper is to describe the clinical presentation, magnetic resonance imaging findings and histopathological and immunophenotypical characteristics of a cerebral cortex ganglioglioma in a 7-year-old Border Collie. The dog presented an acute onset of tonic-clonic epileptic seizures. Magnetic resonance imaging revealed a well-defined large intra-axial mass located on the left forebrain, mainly affecting the frontal cortex. The histopathological examination of the mass revealed a diffuse proliferation of neoplastic glial cells mixed with anomalous neuronal bodies. Immunohistochemical analyses confirmed the presence of two different populations of neoplastic cells. Most neoplastic glial cells were immunoreactive to glial fibrillary acidic protein and the other subset of neoplastic cells were positive to neuronal markers such as PGP 9.5, synaptophysin and neuron-specific enolase, suggestive of neuronal cells. These findings confirmed the diagnosis of a cerebrocortical ganglioglioma. To the authors’ knowledge, this is the first description of a cerebral cortex ganglioglioma in a dog.

**Abstract:**

Gangliogliomas are extremely rare tumors of the nervous system composed of neoplastic glial and neuronal cells. The aim of the present paper is to describe the clinical presentation, magnetic resonance imaging (MRI) findings and histopathological and immunophenotypical characteristics of a cerebral cortex ganglioglioma in a 7-year-old Border Collie. The dog presented an acute onset of tonic-clonic epileptic seizures. MRI revealed a well-defined large intra-axial mass located on the left forebrain, mainly affecting the frontal cortex. Following humane euthanasia, the histopathological examination of the mass revealed a diffuse proliferation of neoplastic glial cells mixed with anomalous neuronal bodies. Immunohistochemical analyses confirmed the presence of two different populations of neoplastic cells. Most neoplastic glial cells were immunoreactive to glial fibrillary acidic protein (GFAP) and the other subset of neoplastic cells were positive to neuronal markers such as PGP 9.5, synaptophysin (SYN) and neuron-specific enolase (NSE), suggestive of neuronal cells. These findings confirmed the diagnosis of a cerebrocortical ganglioglioma. To the authors knowledge, this is the first description of a ganglioglioma of the cerebral cortex in a dog.

## 1. Introduction

Gangliogliomas are slow-growing tumors characterized by a combination of dysplastic neurons and neoplastic glial cells [1,2]. These rare nervous system tumors have been described in humans, in which constituting approximately 0.33–1.3% of primary central nervous system tumors [3]. Gangliogliomas in humans represent the most common tumor entity in children and young adults suffering from chronic focal epilepsies, and most frequently, it affects the temporal lobe followed by the frontal lobe [3,4,5,6]. Immunohistochemical analysis is an essential technique to establish the diagnosis of ganglioglioma.

In veterinary literature, gangliogliomas have been anecdotally described in horses, dogs, hedgehog, rodents and calf [7,8,9,10,11,12,13]. In dogs, only three cases of gangliogliomas have been described, neither of which affected the cerebral cortex [8,10,11]. The sole case with seizure-like neurological signs has been reported in a puppy with ganglioglioma affecting the thalamus [11]. Immunohistochemical analyses were performed in those cases, including glial fibrillary acidic protein (GFAP), neuron-specific enolase (NSE), neurofilament protein and proliferating cell nuclear antigens (PCNA) [8,11]. In this case report, we describe the clinical, diagnostic imaging and histopathological and immunohistochemical findings of a ganglioglioma located at the level of the frontal cerebral cortex in a 7-year-old Border Collie.

## 2. Case Presentation

A 7-year-old entire female Border Collie was referred to the Neurology service of *Centro Clínico Veterinario Indautxu* following an acute onset of cluster seizures. The dog was 2 month old when acquired, and the owners did not report any relevant diseases since then. Very subtle generalized incoordination and difficulty to climb stairs were noted by the owners a few months earlier, but no additional investigations were performed.

A physical exam did not show any relevant abnormalities. A neurological examination revealed an abnormal mental status and non-ambulatory tetraparesis, with decreased postural reactions in all four limbs. A cranial nerve examination showed absent menace response on the right eye. Based on this, neuroanatomical localization for an underlying lesion was the left prosencephalon.

The main differential diagnosis included neoplasia, inflammatory/infectious diseases and vascular conditions. Complete blood count, serum chemistry and urinalysis were unremarkable and thoracic radiographs did not reveal any significant abnormalities. MRI of the brain was performed under general anesthesia.

MRI of the head was performed using a 0.2 T permanent open magnet (Hitachi, Tokyo, Japan, Airis mate). T1-weighted (T1W) and T2-weighted (T2W) images were obtained in transverse, sagittal and dorsal planes. Additionally, T2-weighted fluid attenuation inversion recovery (FLAIR) images were obtained in a transverse plane. T1W images in transverse and sagittal planes were also acquired after intra-venous contrast administration (0.1 mmol/kg bodyweight, gadoterate meglumine [Dotarem^®^; Guerbet, Roissy CdG cedex, France]). The study revealed a large, solid, well-defined, lobulated oval-shaped and intra-axial mass affecting the left forebrain. The lesion measured approximately 4.9 cm rostrocaudal, 2 cm mediolaterally and 3.4 cm dorsoventrally, and extended caudally from the left olfactory bulb along the medial aspect of the left frontal cortical area and cingulum reaching the level of the parietal cortical area, medial to the left lateral ventricle. The mass showed a marked hyperintense signal on T2W, mixed heterogenous hyper to isointense signal on FLAIR and marked hypointense signal on T1W, compared to the gray matter; no evident contrast enhancement was identified. The mass severely compressed the surrounding brain parenchyma; it caused ventral displacement of the corpus callosum, partial collapse of the rostral aspect of the left lateral ventricle, attenuation of the cerebral sulci, midline shift to the right with subfalcine herniation and transtentorial herniation (Figure 1). The lateral ventricles were moderately distended by normointense cerebrospinal fluid due to the mass’ compression of the interventricular foramina/third ventricle.

The main differential diagnosis included primary brain neoplasia, being glioma most likely. Others secondary brain tumors such as round-cell neoplasia (lymphoma, histiocytic sarcoma) were considered unlikely. Inflammatory (meningoencephalitis of unknown origin) or infectious diseases were also considered less likely. Cerebrospinal fluid tape was not performed due to the increased intracranial pressure and the associated risks.

Given the presumptive diagnosis of a brain tumor and the poor prognosis, the dog was humanely euthanized. A complete necropsy was performed and the brain was submitted for a histopathological analysis, previously fixed in 10% buffered formalin for more than 24 h. Immunohistochemistry (IHC) was performed to characterize all cell types and in Table 1, we detail the IHC characteristics. For IHC, 3 μm formalin-fixed and paraffin wax-embedded sections of tissue were dewaxed and rehydrated, followed by antigen retrieval in a steamer prior to IHC performed manually. Detection of bound primary antibody was achieved using the Dako EnVision + System-HRP kit, with DAB as the chromogen. Pre-treatment in all IHC was HIER citrate buffer 0.01 M pH 6.

Grossly, the brain showed an enlarged and softer left cerebral hemisphere compared with the right one. On transverse section (Figure 2A), a focal intra-axial space-occupying lesion was present on the left cerebral hemisphere, extending from the dorsomedial part of the left frontal till the parietal cerebral cortical areas (or cortices), invading subjacent cingulate gyrus and even reaching the septal area. It was ill-defined with a gelatinous consistence and a grey/beige coloration. It was accompanied by an enlargement of the adjacent white matter (*Corona radiata* and *Corpus callosum*).

On microscopic examination, the mass corresponded to a diffuse neoplastic proliferation originated on the grey and white matter transition zone, showing a marked lax growth pattern, and infiltrating diffusely the adjacent nervous parenchyma (Figure 2B). The predominantly neoplastic cells showed astrocytic features. They were irregular to star-like, middle-sized cells showing a round or irregular pale nuclei with laxed chromatin and eosinophilic cytoplasm with numerous fibrillary processes; bi- or multinucleated cells were often observed (Figure 2D and Figure 3A). Cell pleomorphism and anisokaryosis were evident. In some areas, these cells increased in size due to the accumulation of an eosinophilic cytoplasm with an eccentric unique or double nuclei and evident cytoplasmic processes (gemistocytic astrocytes) (Figure 2C,D). No mitotic figures were evident. A second cell population was detected intermingled with the previously described. It consisted of numerous, isolated, neuronal-like cell bodies showing anomalous morphological nuclear features (double nucleus and irregular shaped, nuclear invaginations) (Figure 3B–D) and varying amounts of Nissl substance. No mitoses were detected among this neuronal cell population. In addition, capillary vessels showed a proliferative pattern (Figure 2C).

Most neoplastic cells were immunoreactive to GFAP (Figure 3A,B). The other subset of neoplastic cells had neuronal origin confirmed by their immunopositivity against neuronal markers such as Neuron specific enolase (NSE), synaptophysin (SYN) and the protein gene product 9.5 (PgP 9.5) (Figure 3C,D). SYN allowed visualization of binuclear and dysplastic neurons with intranuclear neuronal invaginations. (Figure 3D). These cells did not show immunopositivity for NeuN. Scarce intermixed cells mainly showing oligodendrocyte morphology resulted immunopositive for oligodendroglial marker (Olig-2).

These findings confirmed a mixture of atypic astrocytic glioma, including immature dystrophic neuronal cells being consistent with a grade II cerebral ganglioglioma, a rare type of primary nervous system mixed tumor.

## 3. Discussion

According to the World Health Organization (WHO) Classification of human tumors of the Central Nervous System, gangliogliomas are classified in the group of the glioneuronal and neuronal tumors [14]. No key genes or proteins have been identified for the pathogenesis and diagnosis of this type of tumor and the origin of gangliogliomas remains unknown despite multiple hypothesis being proposed. Tumors consisting of neuronal cells are usually divided into those of a more primitive type, including medulloblastoma and neuroblastoma, and those of an adult type, including ganglioneuroma and ganglioglioma.

Gangliogliomas are extremely rare tumors in dogs; only a few focal cases have been reported in the thalamus of a puppy [11], in the pituitary gland of a mature dog [8] and an unusual intraocular location [10]. To the best knowledge of the authors, this is the first description of a ganglioglioma in the cerebral cortex of a dog. Multifocal ganglioglioma has been previously reported in the piriform and temporal lobe of a horse, as well as in the mesencephalon and hippocampus [7]. In the present case, the main clinical sign described were seizures, similar to the case of the 4-month-old dachshund with thalamic affection and the horse. In addition, absent menace response was observed in the present case due to the affection of the fontal cortical region, which is involved in the motor pathway of the menace response. Motor facial dysfunction with normal cranial nerve reflexes was also evident in the horse case. Two other cases of gangliogliomas in veterinary medicine were identified in the spinal cord of a calf and in a European hedgehog presented with ataxia, paraplegia and urinary bladder dysfunction [9,12].

The tumor’s predilection site in humans is the temporal and frontal lobes, resulting in intractable seizures in the majority of patients [3,4,5,6]. Developing seizures can be complex partial seizures or generalized tonic-clonic seizures. Other clinical manifestations are headache, alteration in consciousness or focal neurologic deficits depending on tumor location [4]. Similar to human cases, our dog displayed generalized epileptic tonic-clonic seizures. Outcome of antiepileptic treatment and intractability of the seizures in the present case could not be evaluated due to the prompt euthanasia.

MRI findings of human gangliogliomas are not well established due to the wide variety of signals, probably due to the cell heterogenicity of these tumors. Gangliogliomas appearance have been described as hypo or isointense on T1W sequences, hyper or hypointense on T2W sequences and with variable contrast enhancement. Quite frequently, this type of tumor shows cystic (hypointense areas on T1W and FLAIR in addition to hyperintense on T2W) and mineralized areas, but can also be found as a solid, poorly defined mass mainly in the temporal lobes [4,6]. In human medicine, gangliogliomas are radiologically classified as cystic masses, cystic-solid and solid [6]. Perilesional edema is another feature present in some cases. Rarely, adjacent skull bone destruction has been described [6]. According to the human medicine literature, the present case must be classified as solid mass with perilesional edema.

Most of the gangliogliomas in humans are well differentiated and classified as grade I, however, anaplastic cases (grade III) have been also described [1,2]. The glial component of each tumor was graded from I/III based on standard histopathological criteria used for astrocytomas: the presence or absence of nuclear atypia (hyperchromatism or pleiomorphism), mitotic activity, vascular hyperplasia, or necrosis. In the present case, atypical and dystrophic cellular features observed in both cell populations together with immunohistochemical results moved us to classify it a as grade II tumor with a worse prognosis. In the two puppies, the histopathology also revealed mineralized deposits [10,11]. Mineral deposits and dystrophic calcification were other imaging and histopathological findings in humans with slow-growing neuroglial tumors as gangliogliomas [1,2,4,6]. In the present case, no mineral deposits were evident.

Immunohistochemical techniques are essential to establish the diagnosis of ganglioglioma. In the present case, we identified both cellular components of the tumor: astrocytes (immunopositives for GFAP) and neuronal cell bodies (immunopositives for SYN, NSE, PgP 9.5). In the present case, few cells resulted immunopositives for Olig-2, a marker for neural progenitors and oligodendrocytes, and no immunopositivity was observed against NeuN, a marker of mature neurons [15]. These results probably indicate an immature origin for the neuronal population of this tumor. Recently, two cases of oligodendroglioma with neuronal differentiation have been published in two boxer dogs [16]. These tumors were considered to resemble human ‘oligodendroglioma with ganglioglioma-like maturation’. In the present case, neither morphological nor immunohistochemical features of oligodendroglioma were present. Clinical progression of gangliogliomas is usually protracted and progressive in humans. Due to the slow-growth nature of this tumor, most of the patients have a mean duration of clinical signs between 6 and 9 years before the diagnosis, especially in gangliogliomas with cerebral location [4,5,17]. The outcome of human gangliogliomas is reasonably good compared with other types of gliomas. The tumor location is the only predictive variable in one study, giving poorer outcome to spinal cord or brainstem location compared with cerebral location [17]. The recent development of disruptive high-resolution microscopies contributes to improving the knowledge related to the morphological properties of those cells involved in health and disease. In this framework, epi-fluorescence imaging [18] and sophisticated volumetric assays by atomic force microscopy [19] could be employed as a promising tool in the characterization of gangliogliomas. The treatment of choice in humans is total surgical resection based on the slow-growing and non invasive nature of this tumor [4,5]. The majority of patients with epilepsy treated surgically had a complete and significant reduction of seizure frequency. Depending on the study, the recurrence survival rate was 95% and 97% at 5- or 7.5-years follow-up, respectively [5,17]. The role of adjuvant radiotherapy and/or chemotherapy is controversial and most studies show no significant benefit [4,5,17]. Histological grade was not predictive of outcome, although higher-grade tumors tend to have a shorter time to recurrence. Lower rates of recurrence were found in patients with tumors classified as WHO Grade I lesions, patients with temporal lesions, patients who underwent complete tumor resection and patients with long-standing epilepsy [5]. High variability in the progression rate of clinical signs has been observed in veterinary literature. Protracted one-year progressive disease was described in the case of the horse, however, in the canine cases and in the present case, the clinical signs were quickly progressive [7,10,11]. Little is known about the outcome and survival rate after surgical resection in veterinary cases, due to the scarce number of cases. In extra-axial gangliogliomas (pituitary and intraocular location), both patients experienced complete resolutions of clinical signs after transsphenoidal hypophysectomy or eyeball enucleation [8,10]. The rest of the intracranial or intramedular gangliogliomas were promptly euthanized without any treatment established [7,9,11,12].

## 4. Conclusions

To the authors knowledge, this is the first report of intracranial ganglioglioma located in the cerebral cortex in a dog. Despite ganglioglioma is a rare neoplasia, it must be considered in the differential diagnosis of a solitary solid mass on the cerebral cortex of young or adult dogs. Biopsy or postmortem histopathology and immunohistochemical evaluation remain essential to achieve the correct diagnosis and must be crucial to give accurate prognosis.

## Figures and Tables

**Figure 1 vetsci-09-00514-f001:**
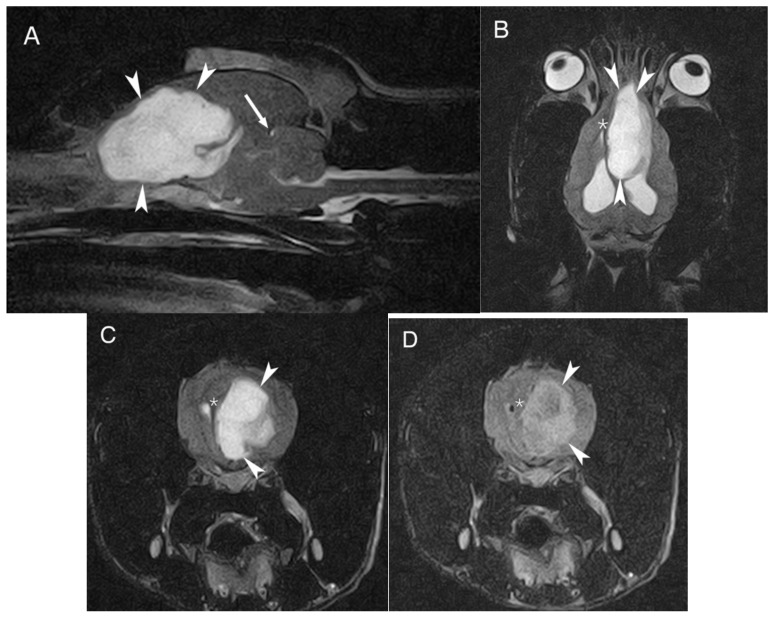
MRI of the dog´s brain. Sagittal T2W (**A**), dorsal T2W (**B**) and transverse plane on T2W (**C**) and FLAIR (**D**) at the level of the frontal cortical area showing the large intra-axial mass (arrowheads) located on the left forebrain, hyperintense on T2W and FLAIR. The asterisk shows the midline shift to the right and the arrow shows the transtentorial herniation.

**Figure 2 vetsci-09-00514-f002:**
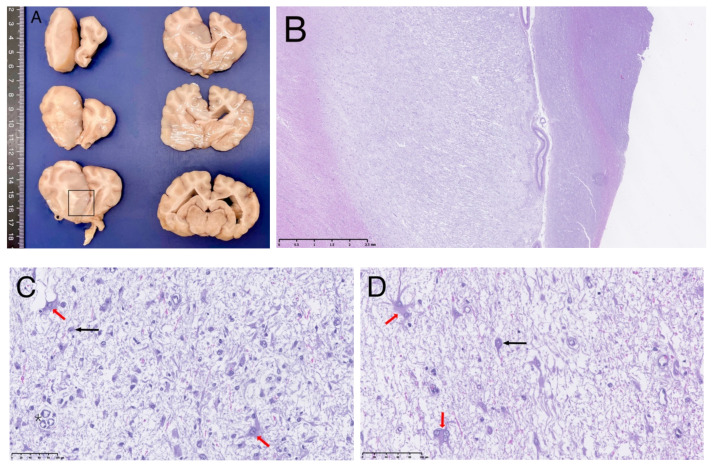
Transverse sections of fixed brain at different levels (**A**) and corresponding microscopic examination on HE of the abnormal area (**B**–**D**). (**B**) is a sub-gross image as shown by the scale bar at 2.5 mm and (**C**,**D**) show microscopical images (scale bar at 100 μm). In (**A**), left cerebral hemisphere shows a large focal intra-axial on its medial part, causing marked midline shift to the right. The square indicates the area where microscopic figures were obtained. In (**B**), a pale and disorganized cerebral cortical parenchyma is evident, corresponding to the diffuse neoplastic proliferation. (**C**) shows activated astrocytes (red arrow) mixed with neuronal bodies (black arrow in the image) and vascular proliferation (asterisk in the image). At higher magnification, (**D**) dystrophic astrocytes showing bi- or multinuclei (red arrows) are present mixed with small pyramidal neuronal bodies (black arrows).

**Figure 3 vetsci-09-00514-f003:**
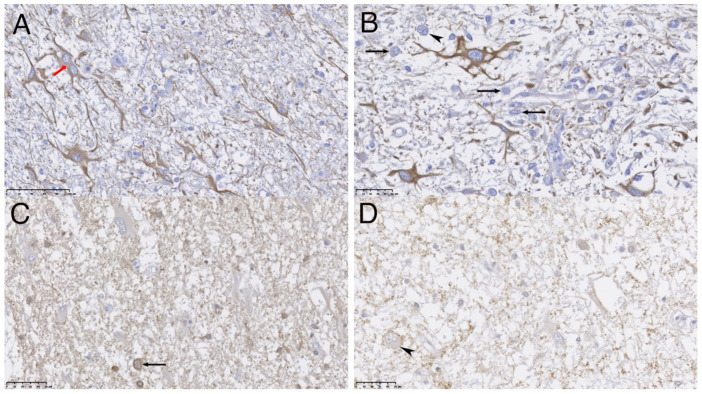
Immunohistochemistry (IHC) analyses of the nervous tissue, scale bar show the correspondence of 100 μm (**A**) and 50 μm (**B**–**D**). Immune positive cells are stained in brown. (**A**,**B**) show activated and dystrophic astrocytes immunopositive to GFAP (red arrow), one showing a double nucleus (red arrow) mixed with neuronal cell bodies (immunonegatives and counterstained with hematoxylin, black arrows) also dystrophic, one of them showing double nucleus (marked with black arrowhead). (**C**) shows small immune stained neuronal cell bodies and corresponding neuropile processes to PgP 9.5, mixed with immune negative large astrocytes; a binucleated neuronal body is present (black arrow). (**D**) shows granular surface SYN immunopositivity on neuronal bodies and processes surface, and surrounding neuropil; a large dysplastic binucleated neuronal body is present (marked with black arrowhead).

**Table 1 vetsci-09-00514-t001:** Antibodies characteristics employed on brain tissue.

Antibody	Source	Host Species	Dilution
GFAP ^1^	Agilent DAKO, Z0334, CA, USA	Polyclonal Rabbit	1/5000
PgP 9.5 ^2^	UltraClone, RA 95101, Isle of Wight, U.K.	Polyclonal Rabbit	1/1000
Olig—2 ^3^	Merck KgaA, Darmstadt, Germany	Polyclonal Rabbit	1/500
NSE ^4^	Agilent DAKO, M0873, CA, USA	Monoclonal Mouse	1/600
SYN ^5^	Agilent DAKO, A0010, CA, USA	Polyclonal Rabbit	1/100
NeuN ^6^	Merck KgaA, MAB 377, Darmstadt, Germany	Monoclonal Mouse	1/300

^1^ GFAP, glial fibrillary acidic protein; ^2^ PgP 9.5, protein gene product 9.5; ^3^ Olig-2, oligodendrocyte transcription factor; ^4^ NSE, neuron-specific enolase; ^5^ SYN, synaptophysin; ^6^ NeuN, neuronal nuclear antigen.

## Data Availability

Not applicable.

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
