# Peer review of "First Case of a Cerebrocortical Ganglioglioma in a Dog"

_vetsci, 2022, doi:10.3390/vetsci9100514_

Round 1

Reviewer 1 Report

The manuscript titled “FIRST CASE OF A CEREBROCORTICAL GANGLIOGLIOMA IN A DOG” by Martín-Muñiz, L.; et al. is a case report work where the authors deliver the unprecedented observation of cerebrocortical ganglioglioma in dogs. Even, if it exists previously studies where gangliogliomas were found in other cranial regions like thalamus (Reference number 11), no earlier scientific works are found in literature with the outcomes shown in the present manuscript. For this reason, this work could have a strong future impact in this academic field. The authors have combined numerous complementary techniques like magnetic resonance imaging and immunophenotypical and histopathological assays in order to unambiguously assess the tissue from the studied dog. The gathered findings may be relevant for the prognosis of cancer diseases in animals, specially dogs. The results achieved are well-discussed during the main body of the reported manuscript. The scientific paper is well written. In my opinion the present manuscript is innovative and the methodological approached used matches with the scope of Veterinary Sciences journal. For the above described reasons, I recommend the publication in Veterinary Sciences once the following remarks will be fixed:

--------

TITLE

Please, authors should write the title in lowercase letters.

--------

NAMES AND AFFILIATIONS

It lacks this information. Please, authors should add these details.

--------

ABSTRACT

Abstract is clear, concise and fully informative. Not changes are required in this section.

--------

KEYWORDS

Authors should consider to add the term “ganglioglioma” in the Keywords list.

--------

INTRODUCTION

“These rare nervous system tumor have been (…)” (line 23). Please, change the aforementioned sentence by “These rare nervous system tumors have been (…)”.

“~0.33-1.3%” (line 23). Authors should take care of the significant figures and homogenize them through the entire manuscript body text.

“In veterinary literature gangliogliomas (…)” (line 28). It lacks a comma: “In veterinary literature, gangliogliomas (…)”.

“Including GFAP, NSE, neurofilament protein and proliferating cell nuclear antigens (PCNA) [8,11].” (lines 32-33). Authors should define the abbreviations “GFAP” and “NSE” even if both of them have been previously stated in the Abstract section.

--------

CASE PRESENTATION

This section is accurate and well-written. Authors should add scale bars in those images from Figure 1 (line 74). Moreover, the scale bar information details from the previous Figure 1 in addition to Figure 2 (line 113) and Figure 3 (line 143) should be also appeared in the respective Figure caption.

--------

DISCUSSION

In general terms, the article is well-discussed and the cited references are appropriate. The following aspect should be further discussed:

I may consider appropriate the introduction of disruptive high-resolution microscopies as promising future avenues to address the morphological properties of those cells involved in health and disease. In this framework, epi-fluorescence imaging [1] or the recently sophisticated developed volumetric assays [2] by atomic force microscopy (AFM) which could be fully employed in the characterization of gangliogliomes [3] have emerged as promising tools in this field. 

[1] Mishchenko, T.A.; Balalaeva, I.V.; Klimenko, M.O.; Brilkina, A.A.; Peskova, N.N.; Guryev, E.L.; Krysko, D.V.; Vedunova, M.V. Far-Red Fluorescent Murine Glioma Model for Accurate Assessment of Brain Tumor Progression. Cancers 2022, 14, 3822. https://doi.org/10.3390/cancers14153822.

[2] Marcuello, C.; Frempong, G.A.; Balsera, M.; Medina, M.; Lostao, A. Atomic Force Microscopy to Elicit Conformational Transitions of Ferredoxin-Dependent Flavin Thioredoxin Reductases. Antioxidants 2021, 10, 1437. https://doi.org/10.3390/antiox10091437.

[3] Au, N.P.B.; Fang, Y.; Xi, N.; Lai, K.W.C.; Ma, C.H.E. Probing for chemotherapy-induced peripheral neuropathy in live dorsal root ganglion neurons with atomic force microscopy. Nanomedicine 2014, 10, 1323-1333. https://doi.org/j.nano.2014.03.002.

--------

BIBLIOGRAPHY

The references are in the proper format of Veterinary Sciences journal. No further improvements are requested in this section.

--------

OVERVIEW AND FINAL COMMENTS

The submitted work is well-designed and the gathered results are interesting for the high-throughput screening of cerebrocortical gangliogliomas in dogs. This first observation could pave the way to future studies in order to better understand the mechanisms related to this type of disease not only in dogs, but also in other living organisms. For this reason, I will recommend the present scientific manuscript for further publication in Veterinary Sciences once all the aforementioned suggestions will be properly fixed.

Reviewer 2 Report

The manuscript focuses a case of ganglioglioma which is an exceedingly rare tumor and, to the authors knowledge, the present report represents the first description of a ganglioglioma of the cerebral cortex in a dog.

The case is interesting, and the manuscript is very well written. The introductive part is exhaustive, the case has been studied both histologically and immunohistochemically and gave interesting and convincing results which are well commented in the discussion.

I suggest only some extra-minor corrections:

Line 21: delete “a” before “slow growing”

Line 65: change as “The mass severely compressed the surrounding brain parenchyma

Line 105: “comparing” or “compared”

Line 179 and following: avoid to use “our” and use “the present case”

Line 195- 96: ganglioglioma imaging appearance…(delete “imaged”)

Reviewer 3 Report

This is a novel study about cerebrocortical tumours in dog. It's well prepared and documented report. I've only minor comments. The manuscript requires extensive English correction and please explain the abbreviations from line 33.
